# BUDGET-CONSTRAINED ACTIVE LEARNING TO DE-CENSOR SURVIVAL DATA

## ABSTRACT

Standard supervised learners attempt to learn a model from a **labeled** dataset. Given a small set of labeled instances, and a pool of unlabeled instances, a **budgeted learner** can use its given budget to pay to acquire the labels of some unlabeled instances, which it can then use to produce a model. Here, we explore budgeted learning in the context of survival datasets, which include (right) censored instances, where we know only a lower bound $c_i$ on that instance's time-to-event $t_i$. Here, that learner can pay to (partially) label a censored instance – *e.g.*, to acquire the actual time $t_i$ for an instance [*e.g.*, go from (3yr, censor) to (7.2yr, uncensored)], or other variants [eg, learn about 1 more year, so go from (3yr, censor) to either (3.2yr, uncensored) or (4yr, censor)]. This serves as a model of real world data collection, where follow-up with censored patients does not always lead to uncensoring, and how much information is given to the learner model during data collection is a function of the budget and the nature of the data itself. Many fields – such as medicine, finance, and engineering – contain survival datasets with a large number of censored instance, and also operate under budget constraints with respect to the learning process, thus making it important to be able to apply this budgeted learning approach. Despite this importance, very few other projects have explored this. We provide both experimental and theoretical results for how to apply state-of-the-art budgeted learning algorithms to survival data and the respective limitations that exist in doing so. Our approach provides bounds and time complexity asymptotically equivalent to [m:]the standard active learning method[m:s]BatchBALD. Moreover, empirical analysis on several survival tasks show that our model performs better than other potential approaches on several benchmarks.

## 1  INTRODUCTION

Often[m: times,] the success of a model is more dependent on the data the model has access to than the quality of the analysis itself. Data plays a large role in the bias, variance, speed of training, and generalizability of the model (Domingos, 2012). However, many factors often come into play that limit the data the model has access to. One such factor [m: exists in]is the cost of obtaining diverse and relevant data that is needed for the model to generalize effectively (Domingos, 2012). Some fields such as medicine which depends on clinical trials can often[m: have trouble getting]with diversity within the datasets, as well as gathering instances can be costly. [m: We wish to build]Our goal is to develop a method to select the most informative instances to learn about in medical and clinical settings while taking the budget available into consideration. For example, if we have a dataset of 50 labeled instances, and a budget to learn about 10 other instances (from a large pool of unlabeled instances), we want to use the given 50 instances to identify the most informative 10.

Here, we want to extend these ideas to *survival datasets* – where the goal is a model that can predict the "time to event" (typically death) for a novel instance. This resembles standard regression, except the training dataset often includes *censored* data points, where the exact time to event is unknown. Instead, these instances typically provide only a lower bound (right-censored) for the time to event. For example, while a cancer study may attempt to follow patients until their death, some patients may leave the study early, or may survive until the end of the study. Here, we would consider them right censored as we do not have the true time of death. Censored data points can be incredibly common in many settings, especially those involving clinical trials (Moghaddam et al., 2022).

Many researchers are uncertain about how to handle such data, meaning some will just remove it. This is a serious issue in a field that needs to use as much data as it can get. Although there are methods for learning models from such survival data, very few methods have been developed for active learning with survival data (Dedja et al., 2023), and so far **no** methods have been developed for budgeted learning with survival data. That is the focus of this work.

The field of *Active Learning (AL)* aims to identify the most important data to learn about (often to obtain the labels of) in order to acquire an effective model, at minimal cost (Ren et al., 2021). Its goal is to minimize the amount of data needed to reach a target accuracy while also enabling the model to achieve higher accuracy as quickly as possible, saving time and compute (less compute is needed to train the model because fewer samples are included in the training data) (Ren et al., 2021).

AL has made significant strides as a field, yet its definition remains vague. Researchers often train models until "convergence", but this term lacks a clear definition, as it relies on arbitrary thresholds. Typically, convergence is declared when the loss changes minimally ($\epsilon$) over a certain number of steps (patience) (Ren et al., 2021). This imprecision arises because both $\epsilon$ and patience are heuristically chosen and can vary by task and model, resulting in an inconsistent understanding of true convergence.

*Budgeted Learning (BL)* tackles this issue by selecting data instances within a budget to minimize model loss, offering a more precise framework for active learning (Lizotte et al., 2003). Budgeted learning attempts to choose the best instances for the model given a budget constraint. If all data instances had the same cost and the task was to minimize the budget rather than working within a predefined one, we would essentially have AL. Budgeted learning is more attuned to real world problems where a predefined budget is often given prior to data collection. We show that once this problem is solved, one can easily also account for real-world situations where data instances may potentially cost different amounts.

Traditional AL makes several assumptions that often don't hold in real-world scenarios. We have already noted that it does not account for prior budget constraints, operating under the premise that the model can run until convergence. Another assumption is that when a data instance is selected for learning, its true label is revealed—an assumption that doesn't always apply in survival analysis. If Alice is censored at time $t = 3$ years, and you run an additional study to learn 1 more year about Alice, her death may occur in that 1 year (*e.g.*, uncensored at $t = 3.2$), or she may be censored at $t = 4$ years. AL often assumes that the full label information is provided when queried. Finally, AL often assumes that if a data instance in the pool does not yet have its true label, it can still be queried to reveal the label. This is again not the case in the real world. Imagine we are studying Alice to track her time of death from cancer. If, during the study, she dies from an unrelated cause — such as being hit by a bus — her time of death from cancer becomes unknown, and we are no longer able to gather more information about her. In this case, Alice represents a censored instance that is also *unqueryable* (meaning no further information can be learned). We also discuss how to adapt our approach to handle situations where different instances have different costs.

**Significance of Settings:** The settings explored in this work encompass a broad range of applications, showing the versatility of our method in handling various survival datasets, instance costs, and scenarios with partial information. While we focus on the classical example of clinical trials, where random right censoring is a common observational scheme and extending the study time may not yield additional information, our approach is equally relevant to other domains. For instance, in industrial reliability studies Ma & Survival (2008), [R2:]where Type I or Type II censoring is frequently employed, extending the study duration can reveal more events and improve insights. [R4:]Additionally, our method has potential applications in the financial sector Gao & He (2020), such as in credit risk modeling or portfolio survival analysis, where understanding time-to-event data under cost constraints is critical.

**Theoretical Results**: [R2:] [We have addressed our contributions formally below.] 1. We reduce the problem given to an instance of the maximum coverage problem by creating a modified version of the

BatchBALD (Kirsch et al., 2019) we call $BB_{surv}$, so that it works in surival settings and with partial information. 2. We show the problem to be NP-Hard in nature but also present the known greedy algorithm recognized as the optimal approximation algorithm unless $P = NP$. This algorithm achieves a guaranteed lower bound of $(1 - 1/e)$ of the optimal information content possible, also shown in the BatchBALD paper. 3. Finally, we provide a new greedy algorithm that achieves the same guaranteed lower bound even if the cost of acquiring data instances are not uniform. This is primarily done by extending the problem to the **budgeted** maximum coverage problem.

**Experimental Results**: For the experimental results the work provides the following contributions: 1. we compose a series of algorithms to compete with ours in this setting. Since this setting has never been done before to our knowledge, we needed to create other algorithms to test our method against. The algorithms we created involve controls such as random and 3 "sanity-check" algorithms which are benchmark strategies people might try within these settings. We also slightly modify BatchBald and two other well known AL algorithms to work within these settings. 2. we evaluate our acquisition function against the rest on 3 real world survival datasets primarily using the MAE-PO (Qi et al., 2023a) evalution method, however we discuss and present other metrics in Additional Data B. 3. We demonstrate that our version of BatchBald refered to as $BB_{Surv}$ performs significantly better than other methods in our settings.

## 2 RELATED WORK

### 2.1 BUDGETED LEARNING

Although a relatively new field, several studies have explored its application in various scenarios (Kapoor & Greiner, 2005a;b; Khan & Greiner, 2014). While these papers introduce the budgeted learning field and apply it to a variety of tasks, none have applied budgeted learning to survival data where it is possible to (partially) "decensor" the labels.

### 2.2 INCORPORATING ACTIVE LEARNING WITH SURVIVAL ANALYSIS

There are very few works in the intersection of AL and survival analysis. Vinzamuri et al. (2014a;b) employ a semi-parametric deep learning model based on the Cox Proportional-Hazards framework, which restricts its applicability to fewer datasets. Furthermore, neither these papers, nor Nezhad et al. (2019), incorporate budget constraints or consider the incremental decensoring of labels.

Hüttel et al. (2024)[R4:] extends the BALD framework to right-censored data, whereas our work employs the more mathematically complex BatchBALD architecture. Additionally, our method adjusts the BALD framework differently to incorporate budget constraints, incremental label updates, and to handle survival analysis with non-uniform instance costs.

Finally, Dedja et al. (2023) propose a AL approach on survival data that includes a mechanism for incrementally updating the label via an oracle. However their increments are random, meaning the updated information is not integrated into the acquisition strategy. Furthermore, their method their method is incompatible with deep learning models as it relies on a random forest model which has significant limitations when scaling to larger datasets. Additionally, they do not address scenarios where the costs of instances are unique.

## 3 FORMULATION OF THE PROBLEM

In traditional AL schemes, the learner starts with some labeled instances and many unlabeled instances, queries an oracle for label information about some instances, then repeats. Here we do not need the process to repeat as we only query the Oracle one time and then evaluate the performance

of the model. Furthermore we make important changes to the formulation of the traditional AL problem to account for the survival data and incremental updates.

Given a dataset $D = \{x_i, y_i^t, y_i^e\}_{i=1}^n$, [R2:] $x_i$ represents the covariates of the dataset, $y_i^t$ denotes the event time for each data instance. and $y_i^e$ denotes the censored value for each instance. We can divide this [m: randomly] into a test and training set: $D_{\text{train}} = \{x_{A,i}, y_{A,i}^t, y_{A,i}^e\}_{i=1}^L$ and $D_{\text{test}} = \{x_{B,i}, y_{B,i}^t, y_{B,i}^e\}_{i=1}^{n-L}$. Where L is the size of the training data. [R2:] For the training data we further censor the $y_{A,i}$ values so we can see the effects of querying and thus the training data also has censored times and event values (0 if censored and 1 if not), $\{c_i^t, c_i^e\}_{i=1}^L$. [R2:] $y_{A,i}^t$ and $y_{A,i}^e$ now represent what the Oracle knows while $c_i^t$ and $c_i^e$ represent what the learner is given. We can then query the oracle, which will update $c_i^t$ and $c_i^e$ based on $y_{A,i}^t$ and $y_{A,i}^e$. [R3:] [changed the notation as per R3 request; and also changed from $data\_train$ to $L$]. Thus the training dataset is actually split into $D_{\text{train}}^{oracle} = \{(x_{A,i}, y_{A,i}^t, y_{A,i}^e\}_{i=1}^L$ which is what the oracle sees and $D_{\text{train}}^{learn} = \{(x_{A,i}, c_i^t, c_i^e\}_{i=1}^L$ which is what the learner sees. From now on we will use $D_{train}$ to refer to $D_{train}^{learn}$. [R2:] [We have made the notations more detailed to explicitly show what information the learner and the oracle see.]

The $i^{th} data point comes with a given cost c_i$ , where each data point has the same cost in the *uniform* setting and can be different in the *non-uniform* setting. A budget $B$ is provided for a query as well. In a query, we can choose a batch of data instances via use of some *acquisition function*, which is used to evaluate the value of a batch of instances, such that the sum of the costs of the instances chosen is less than or equal to $B$.

We wish to find an acquisition function that within the confines of the budget tells us which data instances we should "decensor". But how we are allowed to "decensor" or gain information about the data instances depends on how the information is gathered. If for example a study is done on a queried instance, if the study is an [R2:] $\mathbb{I}$ [Changed the font of I, as requested] year study (for example, $\mathbb{I} = 10$ refers to a 10-year study) then you only learn $\mathbb{I}$ more years about the instance (so an instance might go from being censored at 5 years, to being censored at 8 years). We consider this formulation in our work. Notice that this is a generalization over traditional AL methods, which often allow one to know the exact time of event after a query, which is equivalent to setting $\mathbb{I} = \infty$.

To define this notion of "decensoring" rigorously, we introduce an $\mathbb{I}$-oracle with $\mathbb{I} \in \mathbb{R}^+$. In any query, we can choose a batch $B \subseteq D_{\text{train}}$. For each element $(x_{A,j}, y_{A,j}^t, y_{A,j}^e, c_{A,j}^t, c_{A,j}^e) \subseteq B$, the $\mathbb{I}$-oracle updates the values as follows: $c_{A,j}^t = \min(c_{A,j}^t + I, y_{A,j}^t)$ and $c_{A,j}^e = (1_{c_{A,j}^t = y_{A,j}^t}) * y_{A,j}^e$.

Now we wish to define the model that will be doing the learning in this protocol. We have a Bayesian model $M$ where the model parameters $\omega$ follow the distribution $p(\omega \mid D_{\text{train}})$. For a given data point $x$ and a classification outcome $y \in \{1, \ldots, t\}$, the model's predictions are represented by $p(y \mid x, \omega, D_{\text{train}})$. The dependence of $\omega$ on $D_{\text{train}}$ signifies that the model has been trained on the dataset $D_{\text{train}}$. [R2:] In survival analysis, time is often continuous but can be discretized into time bins for this formulation. Many active learning algorithms rely on discrete labels, and using a large number of bins can effectively approximate continuous time. We found that this simplification yields good results even without a large number of bins.

[R3:] Finally we will use the MAE-PO measure defined in Qi et al. (2023a). That paper compares this metric against many others, to show that it is an effective metric closely related to the well-known mean absolute error (MAE) metric in regression. We also evaluate using other more traditional metrics as well, however as argued in Qi et al. (2023a), the traditional metrics fail in their interoperability and often provide a less useful evaluation of a survival model.

**Initial Assumptions:** We wish to make explicit here the exact assumptions we are making in this formulation. [m:] Assumptions 1 and 2 are assumptions novel to this problem, however assumption 3 is often made in survival settings. [R3:] [As recommended we made our assumptions numbered]1. The value of $\mathbb{I}$ does not change throughout. [m:] This assumption is for simplifying purposes of the problem and experiments, however, our formulation does not demand this be true. 2. Instances can be queried only once per query. Since the oracle may not give the full information in one query, it does make sense to allow the querying of the same point multiple times. We decided to create this assumption however to simplify the code and the experimental settings. However, the theoretical work for generalizing over this assumption is shown in the Theoretical Analysis section A. A final

note about this assumption is that if in a medical setting there is a follow up study, oftentimes that study's length cannot change, *m:* ~~thus it makes no sense to ask for two of the same instance in the same query if there will be only one follow up study~~therefore, requesting the same instance twice in a single query is illogical if only one follow-up study will be conducted. The appendix will discuss some real world scenarios where this assumption does not hold, however for many medical situations, this assumption is realistic *m:* ~~to expect~~. 3. The final assumption we have is that censoring is independent of the features, and is done uniformly for each instance. This is a common assumption made in the field of survival, *m:* ~~however work has been done in generalizing over this assumption that would be a good place to do future work in this area~~However, efforts to generalize this assumption have been made, providing a promising direction for future research in this area (Qi et al., 2023a). *R3:* This assumption can also be observed in many real-life scenarios, for example, in clinical trials where patients drop out of the study for reasons unrelated to their health condition or treatment efficacy. Such censoring is independent of the features being analyzed, like age, gender, or baseline health metrics, allowing the survival analysis to remain unbiased. Similarly, in reliability studies of mechanical systems, censoring can occur when testing is stopped due to budget constraints or time limits rather than any inherent property of the system.

## 4 OUR METHOD

### 4.1 UTILIZING BATCHBALD FOR MUTUAL INFORMATION ESTIMATION

The BatchBALD algorithm (Kirsch et al., 2019) is a state-of-the-art active learning (AL) method, enabling the computation of mutual information between multiple data instances and model parameters. Mutual information, rooted in information theory, quantifies the amount of information one random variable provides about another. It serves as a key metric for assessing uncertainty by evaluating how much an observation reduces uncertainty about a model's parameters.

Hoffmann & Onnela (2023) demonstrate that in the limit, *m:* ~~minimizing many commonly used uncertainty measures aligns with mutual information in terms of reducing Bayesian risk~~minimizing commonly used uncertainty measures aligns with mutual information in reducing Bayesian risk. Thus, mutual information emerges as a critical metric for this framework. *m:* ~~Moreover, the flexibility of the BatchBALD algorithm allows for adaptation to survival analysis tasks, making it a robust tool that generalizes across various assumptions and scenarios in both budgeted learning and survival settings.~~Moreover, the BatchBALD algorithm is flexible enough to adapt to survival analysis tasks. This makes it a robust tool for various assumptions and scenarios in budgeted learning and survival settings.

Kirsch et al. (2019) define the BatchBALD acquisition function using mutual information as: $a_{BatchBALD}(\{x_{1:b}\}, p(\omega \mid D_{\text{train}})) = I(y_{1:b}; \omega | x_{1:b}, D_{train})$, where $I$ represents the mutual information. Kirsch et al. (2019) further define mutual information between the model parameters and a batch of $b$ data instances as follows:

$$I(y_{1:b}; \omega | x_{1:b}, D_{train}) = H(y_{1:b} | x_{1:b}, D_{train}) - E_{p(\omega | Dtrain, x_{1:b})}[H(y_{1:b} | x_{1:b}, \omega, D_{train})] \quad (1)$$

$H(y_{1:b} | x_{1:b}, D_{train})$ determines the information entropy of the labels of the batches given the features and training data, while $E_{p(\omega | Dtrain, x_{1:b})}[H(y_{1:b} | x_{1:b}, \omega, D_{train})]$ defines the expected labels conditioned on the features, training data, and the model parameters.

Following Kirsch et al. (2019) we choose also to not include conditioning on $x_{1:b}$ and $D_{train}$ for elegance, We can compute *R2:* the right term as

$$E_{p(\omega)}[H(y_{1:b} | \omega)] \quad \approx \quad \frac{1}{k} \sum_{i=1}^{b} \sum_{j=1}^{k} H(y_i | \hat{\omega}_j) \quad (2)$$

*R4:* [We included equation numbers]

Kirsch et al. (2019)*R4:* provide a detailed discussion of this factorization, with a key underlying assumption being that, when conditioned on $\omega$, the indices are treated as independent as their dependencies are captured within the parameters. The final step of Equation 2 estimates the expectation by taking k samples of the model parameters. The paper also shows that we can compute the *R2:* ~~right~~left

term as: [R2:] [we simplified the formulas as we felt they were not integral to the result and the try to explain the terms that do exist]

$$H(y_{1:b}) \approx - \sum_{\hat{y}_{1:b}} \left( \frac{1}{k} \sum_{j=1}^{k} p(\hat{y}_{1:b}|\hat{\omega}_j) \right) \log \left( \frac{1}{k} \sum_{j=1}^{k} p(\hat{y}_{1:b}|\hat{\omega}_j) \right) \tag{3}$$

[m:] ~~The formula assumes that data instances are conditionally independent given the model parameters, as their dependencies are captured within the parameters.~~ [m:] ~~However, this does not imply that the mutual information is necessarily zero.~~ The difference in averaging between the left and right terms of the mutual information means they only cancel out when the model outputs show minimal variation, indicating high confidence and low information gain. When model uncertainty is high, the entropy of the model (left term) increases, while the expected entropy of predictions (right term) decreases, leading to higher mutual information (Kirsch et al., 2019).

[m:] ~~BatchBALD was originally tested on classification problems where mutually exclusive classes exist, we can use it here for the time series task if we divide time into exclusive bins which is done in the Multi Task Logistic Regression (MTLR) model, which is a non-parametric survival model.~~ BatchBALD was originally designed for classification tasks with mutually exclusive classes. However, it can be applied to time-series tasks by dividing time into exclusive bins as mentioned in the formulation of the problem, which is a strategy utilized by the Multi-Task Logistic Regression (MTLR) model, a non-parametric approach to survival analysis. [m:] ~~Furthermore,~~ BatchBALD [m:] also requires multiple predictions given the same feature space, which needs a Bayesian model. Fortunately, Qi et al. (2023b) provides a Bayesian model that can give ensemble outputs that [m:] ~~are required and be~~ can be trained on survival data. We use this model due to its simplicity and [m:] ~~the fact that it has been shown to perform well on the datasets we test. There are other models that could be used in this place, however the work done in this paper does not have a preference over these models as long as they are Bayesian, effective, and can work with survival data~~ its demonstrated strong performance on the datasets we tested. While other models could be used instead, our work does not favor any particular one, as long as they are Bayesian, effective, and compatible with survival data.

BatchBALD cannot be used in our settings without first accounting for the censored nature of the instances, as well as the increment $\mathbb{I}$ of the oracle. In order to take these into account, we generate $p_{final}$ probabilities in section 5 which we can then use in place of $p$ in equations 2 and 3. We introduce our method, $BB_{surv}$, which defines a novel acquisition function, $a_{BB_{surv}}$ [R2:] [We define aBBsurv originally here]. This function is constructed by adapting the BatchBALD acquisition function to utilize the newly computed final probabilities.

## 4.2 MAXIMUM COVERAGE AND GENERALIZING OVER UNIFORM COSTS

Given the BatchBALD implementation mentioned already, we have a submodular (Kirsch et al., 2019) metric attached to each batch of instances. The problem is now [m:] ~~in~~ about choosing the batch that [m:] ~~fits within the budget that~~ maximizes the [m:] ~~amount of this~~ metric [m:] ~~gathered~~ while fitting within the budget. The *maximum coverage problem* [m:] ~~(MC)~~ is a combinatorial problem where you are given a set of elements and a collection of subsets, and the goal is to pick a specified number of these subsets so that the total number of distinct elements covered by the chosen subsets is maximized (Khuller et al., 1999). The *weighted maximum coverage problem* extends this by associating a weight with each element, and the objective is to select subsets such that the total weight of the covered elements is maximized, rather than simply the count of covered elements. In the Theoretical Analysis section A, we further discuss these combinatorial problems and demonstrate that the budgeted learning problem with [m:] ~~all costs being equal~~ uniform costs can be expressed as the weighted maximum coverage problem. Similar reductions are done in other AL works as well (Yehuda et al., 2022). The *budgeted maximum coverage problem* further extends the weighted version by introducing a cost constraint to each set. In this variant, each subset has an associated cost, and the goal is to select subsets such that the total coverage is maximized while keeping the total cost within a given budget. T[m:]~~hus, t~~he budgeted learning scenario where the costs are not [m:] ~~needed to be~~ uniform can reduce to the budgeted maximum coverage problem. Fortunately, (Khuller et al., 1999) [m:] ~~provides a different~~ introduces an alternative greedy algorithm that [m:] ~~manages to meet the desired~~ achieves the same desirable lower bound of $(1-1/e)$ of the optimal solution as the original greedy algorithm. We show this new

greedy method in Algorithm 2 in the Theoretical Analysis section A $^{m:}$ [we mention it before Algorithm 1 but Algorithm 2 comes after... is this ok?].

The only problem here is that Algorithm 2 is very computationally expensive as it involves costly operations in order to meet the theoretical bound. In the Theoretical Analysis section A we argue that we can simplify and closely approximate Algorithm 2 for our settings by only considering the information contained in the mutual information, to the ratio of the mutual information to the cost of the batch. This simplification reduces the complixty from cubic to linear. $^{m:}$ ~~This now not only holds when the costs of the instances are different, but also necessarily holds when they are the same, in which case this algorithm reduces to the traditional greedy algorithm~~ Although designed for non-uniform instance costs, the new greedy algorithm simplifies to the original greedy algorithm when costs are uniform, making it applicable to both uniform and non-uniform settings. Therefore, we will be using this algorithm for selecting the batch that attempt to maximize the mutual information metric. Section 4.1 discussed how to generate mutual information values in these settings using $BB_{surv}$, in this section we discussed how we can use this mutual information along with a modified version of the greedy algorithm from Khuller et al. (1999) to create an acquisition function to select a batch with high mutual information with the model parameters. Algorithm 1 illustrates our novel adaptation of the greedy algorithm, incorporating the proposed acquisition function, $a_{BB_{surv}}$. T$^{m:}$~~hus t~~he time complexity of $a_{BB_{surv}}$ is equivalent to that of the BatchBALD acquisition function $^{m:}$~~, with an additional overhead from the greedy algorithm. For all algorithms (except random), we evaluate the ratio of the uncertainty measure to the instance's budget across all settings~~.

---

**Algorithm 1** $BBSurv$ (1 - 1/e-approximate algorithm)

---

**Require:** Budget $B$, queryable pool $\mathcal{D}$pool, model parameters $p(\boldsymbol{\omega} \mid \mathcal{D}\text{train})$
1: $A \leftarrow \emptyset$
2: $costs \leftarrow 0$
3: **while** $costs < B$ **do**
4:     $n \leftarrow \arg\max_i(a_{BBsurv}(A \cup \{x_i\}, p(\boldsymbol{\omega} \mid \mathcal{D}\text{train}))/c_i)$
5:     $A \leftarrow A \cup \{x_n\}$
6:     $costs \leftarrow costs + c_n$
7: **end while**
8: **Output:** acquisition batch $A$

---

## 5 Adjusting for Survival Data

As mentioned before, since the data we are using contains censored data, we cannot directly use these formulations. We must first adjust the methods to account for the survival data. We can do this simply by changing the Bayesian probabilities so that those below censored time are 0.

The models predictions are given by $p(y \mid x, \omega, D_{\text{train}})$, for a data instance with a given set of covariates $x$ and for all $y \in \{1, \ldots, t\}$. If the instances censored time is $c_b$ then $p(y \mid x, \omega, D_{\text{train}}) = 0$ for all y less than $c_b$ and for all times greater than or equal to $c_b$, the probabilities are normalized to get new probabilities called $p_{cens}$ as follows:

$$p_{cens}(y \mid \omega) = \frac{p(y \mid \omega)}{\sum_{i=c_b}^{t} p(i \mid \omega)} \tag{4}$$

We must also now account for the oracle and its increment's impact on the amount of information we gain. More specifically, the larger the increment, the more information we get from any given data instance. If we consider two instances: Alice and Bob, lets say Alice dies 10 years from now, while Bob dies 3 year from now, whether the increment is 1, 5, or 15 years gives far different amounts of information to us about Alice, but gives the same amount of information to us about Bob. Thus we must change the algorithms to account for this. We can adjust the Bayesian probabilities so that, after the oracle-provided increment is applied, all bins beyond the instance's new time bin are treated as a single "after increment" bin. In other words, bins within the range of the increment remain unchanged, but for the acquisition function, all bins outside this range contribute the same level of information and should therefore be aggregated into a single event.

More specifically, if we have an $\mathbb{I}$-Oracle, the model's predictions are given by $p_{cens}(y \mid x, \omega, D_{\text{train}})$, for a given $x$ and for all $y \in \{1, \ldots, c\}$. If x's censored time is $c_b^t$ then the only relevant classes of $y$ for our acquisition function to consider are those classes within the increment $\mathbb{I}$, all classes outside this increment can be grouped as one class as the oracle provides no information about them. Effectively for out method we only want BatchBALD to see the classes in the increment range, and than one additional class which represents the cumulation of all classes outside that range. For the sake of our acquisition function, this can easily in the code by creating a new probability $p_{final}$ which is equal to $p_{new}$ for all $y < c_b^t + \mathbb{I}$, and equal to 0 for all $y$ greater than than $c_b^t + \mathbb{I}$. Then we can compile the rest into one bin:

$$p_{\text{final}}(\ c_b^t + \mathbb{I} \mid \omega\ ) \quad = \quad \sum_{j = c_b^t + \mathbb{I}}^{c} p_{cens}(\ j \mid \omega\ ) \tag{5}$$

*R4:* [doing this is just a convenience for the code of BBsurv. We could equivalently define a new class as the cumulation of all classes outside the increment interval and set all other class probabilities outside the interval to 0; note that would achieve the same effect.]

This may seem a bit counter intuitive, as grouping multiple probabilities into one bin seems almost like we are losing information. But these methods should only take events that can occur in the next query into account, as in our problem definition we are evaluating directly after the next query. Thus taking events that are unknown even after the next query into account will yield less effective results in this task. For all algorithms apart from our method $BB_{surv}$, we use the $p_{cens}$ probabilities rather than $p_{surv}$; this has not been done before in the AL literature and we wish to compare its affects against controls that do not use it.

# 6 ALTERNATIVE ALGORITHMS

We have not found any existing method that addresses the niche area where budgeted active learning (AL) is applied to censored data. An algorithm designed for this domain must account for the budget, handle the increments provided by the oracle, and be compatible with deep learning models.

In this section, we discuss the methods used to benchmark against $BB_{\text{Surv}}$. Given the extensive work on AL, we include two well-known algorithms from the AL domain and a third in the form of BatchBALD, all of which are modified to handle censored data. Additionally, we incorporate three algorithms as "sanity checks," representing simple methods that are easy to test and conceptualize in these scenarios. Finally, we include the random acquisition function as a control.

## 6.1 COMMON AL ALGORITHMS

**Entropy Sampling:** Entropy sampling is a well-established technique in AL (Ren et al., 2021) that focuses on ~~*m:* selecting data points for which the model's predictions are the most uncertain. The underlying principle of entropy sampling is to~~ maximizing the information gain from newly labeled data by targeting instances where the model's predictive distribution has the highest entropy.

$$H(y \mid x) = -\sum_{i=1}^{S} p_{\text{cens}}(y = c_i \mid x) \log_2 p_{\text{cens}}(y = c_i \mid x) \tag{6}$$

where:

- $S$ is the number of possible classes.
- $p_{cens}(y = c_i \mid x)$ is the probability of the data point $x$ being classified as $c_i$.

*m:* ~~T~~The acquisition function then takes the instances with the largest entropy values until the budget is used up. This method has two notable drawbacks from BatchBald; it does not take the Bayesian nature of the model into account, and it does not measure the entropy of a batch (it only measures entropy of an instance).

**Variance Sampling:** *m:* ~~Similar to entropy sampling, t~~This method involves taking the variance of the values in *m:* ~~$p_{cens}(y|x)$~~the predicted class and *m:* ~~using that metric to choose~~choosing the *m:* ~~highest variance~~ values until the budget runs out.

Given $\{p_1, p_2, \ldots, p_T\}$ as the predicted *m:* ~~probabilities~~classes for a *m:* ~~data point~~instance $x$, the variance Var$(x)$ of the predicted probabilities is calculated as:

$$\text{Var}(p) \;=\; \frac{1}{T}\sum_{i=1}^{T}(p_i - \bar{p})^2 \tag{7}$$

where $\bar{p}$ is the mean predicted class.

## 6.2 SANITY CHECKS

**Closest to Half:** Let $p_i$ denote the predicted probability that the event will occur within the decensored time window for the $i^{th}$ instance. For each instance, we compute its absolute distance from 0.5:

$$d_i \;=\; |p_i - 0.5| \tag{8}$$

The goal is to select instances where this distance $d_i$ is minimized – *i.e.*, where the predicted probability is closest to 0.5. We choose the lowest distances here.

**Mean Closest to Middle:** Let the midpoint of the time range $T$ be $T_{\text{mid}} = \frac{T_{\text{max}}+T_{\text{min}}}{2}$, where $T_{\text{max}}$ and $T_{\text{min}}$ are the maximum and minimum possible survival times, respectively. For each data instance $i$, we calculate the distance to the midpoint:

$$d_i \;=\; |\hat{t}_i - T_{\text{mid}}| \tag{9}$$

The algorithm selects data points with the smallest $d_i$, corresponding to those whose predicted survival times are nearest to the midpoint.

**Using Clusters to form Batches:** This method leverages clustering and censoring measures for instance selection. We use Principal Component Analysis (PCA) to reduce the feature space, followed by K-means clustering to group the data. Clusters with higher average censoring measures are prioritized, and instance selection is guided by proximity to cluster centers while respecting cost constraints.

For each cluster, we calculate the average censoring measure based on the time-to-event data and the censoring status. Let the censoring status for each instance $i$ be represented as $c_i$ (where $c_i = 1$ means the instance is uncensored, and $c_i = 0$ means it is censored). The average censoring measure for a cluster $C_j$ is:

$$\text{Censoring Measure for } C_j = \frac{1}{|C_j|}\sum_{i \in C_j} c_i \tag{10}$$

Clusters with higher average censoring measures represent areas with greater uncertainty or incomplete information. We then calculated proximity as $\text{Proximity}(i, \mu_j) = \|X_{\text{PCA},i} - \mu_j\|$. We choose our batch by selecting the minimum proximity to cost ratio for each instance.

**Random:** This method picks a random data instances to query until the budget is used up. We adjust random so that the probability of random choosing any given instance is proportionate to the reciprocal of the instance cost.

## 7 EXPERIMENTS

For each dataset, we first divide it into training and test sets. We assign the labels to time bins as quantiles of the time to event label. We use the same bins as the training data to assign the labels of the test data. We artificially censor points in the training data so that we can decensor them in our experiments. Hiding information from the model to test its performance is common for uncensored data, however, we can do this for censored data as well. For censored data, we can artificially "further" censor them for this purpose, the only difference is that they cannot be decensored until the time to event.

To artificially censor the dataset, a number of data instances in the training data are further censored at a uniformly random time between 0 and their current "true time" — regardless of whether that time is censored or uncensored. This process assigns each instance a "fake time" after the additional censoring. The true time for each instance is kept hidden from the model (only the oracle can see it). Instances where the fake time is less than the true time are considered *queryable*, meaning the ~~y can provide new information when decensored~~ oracle can provide new information about the instance when queried. On the other hand, instances from which no new information can be gained are *unqueryable*. In the dataset, there is a column called `censored`, which traditionally holds a value of 1 or 0 in standard Active Learning (AL) settings. However, in our setup, this column may also take on the value -1. Specifically, a value of 1 indicates that the time of the event is known, 0 signifies that the time is not yet known but could potentially be determined with further querying, and -1 implies that the time is unknown and cannot be obtained through additional queries (e.g., in the case of a subject being hit by a bus).

At the time of querying, the current fake time is compared to the hidden true time by the oracle and updated accordingly. When an instances time becomes decensored by the oracle, it can still be queryable if the resulting fake time is still less than the true time. But, if the fake time is now equal to the true time, that instance is no longer able to be queried.

We explored 3 real world survival datasets: The Study to Understand Prognoses Preferences Outcomes and Risks of Treatment (SUPPORT; 9,105 patients, censored = 32%, 42 features)(Knaus et al., 1995), Medical Information Mart for Intensive Care ((MIMIC)- IV; n= 38520, censored = 67%, features = 93)(Johnson et al., 2022), and The Northern Alberta Cancer Dataset (NACD; 2402 patients, 53 features, 36% censorship)(Haider et al., 2020). These 3 represent different sized datasets containing various percentages of censored data, complexities for the model to learn, and sizes of features space. We used 5000 epochs with a Bayesian Linear MTLR model with the same initial parameters provided in (Qi et al., 2023b) with a spike and slab prior. For evaluation, Table 1 mentions the MAE-PO (Qi et al., 2023a) metric for evaluating survival models, however ~~in~~ results were obtained for other survival metrics including c-index (Haider et al., 2020), and Brier score as well. More results are shown in the Additional Data section B . We also have results using MAE omitting censored data from the test data (as there was relatively fewer in there). Finally, we have run experiments for a setting where the costs are the same for each instance, and one where they are not. For the latter, we gave each instance a random real value cost between 0.2 and 0.8, the 0.2 and 0.8 were chosen so that the minimum is not too close to 0 and the maximum not too close to 1.

Table 1: Comparison of Acquisition Functions accross Datasets and Time Horizons when Budget = 10. MIMIC, NACD, and SUPPORT are represented by M, N, and S.

| Dataset | BB surv | BatchBALD | Entropy | Var | CtH | CfB | MCtH | Random |
|---|---|---|---|---|---|---|---|---|
| M +5y | **4.23 ± .01** | 4.34 ± .02 | 4.28 ± .01 | 4.28 ± .02 | 4.45 ± .01 | 4.32 ± .02 | **4.23 ± .02** | 4.46 ± .01 |
| M +10y | **4.26 ± .01** | 4.35 ± .02 | 4.28 ± .01 | 4.33 ± .02 | 4.31 ± .01 | 4.27 ± .02 | 4.28 ± .01 | 4.28 ± .02 |
| M +100y | **4.18 ± .02** | **4.18 ± .01** | 4.18 ± .02 | 4.27 ± .01 | 4.27 ± .02 | 4.23 ± .01 | **4.18 ± .02** | **4.18 ± .01** |
| N +5y | **3.63 ± .01** | **3.63 ± .02** | 3.64 ± .01 | 3.66 ± .02 | 3.69 ± .01 | 3.70 ± .02 | 3.81 ± .01 | 3.89 ± .02 |
| N +10y | 3.59 ± .01 | 3.60 ± .02 | 3.61 ± .01 | 3.65 ± .02 | 3.71 ± .01 | 3.77 ± .02 | **3.58 ± .01** | 3.74 ± .02 |
| N +100y | 3.67 ± .01 | **3.66 ± .01** | 3.68 ± .01 | 3.73 ± .02 | 3.73 ± .01 | 3.65 ± .02 | 3.68 ± .01 | 3.70 ± .02 |
| S +5y | **2.09 ± .01** | 2.11 ± .01 | 2.12 ± .02 | 2.10 ± .01 | 2.10 ± .02 | 2.10 ± .01 | 2.11 ± .01 | 2.12 ± .02 |
| S +10y | **2.09 ± .01** | 2.09 ± .02 | 2.10 ± .01 | 2.09 ± .01 | 2.09 ± .02 | 2.10 ± .01 | 2.09 ± .01 | 2.11 ± .02 |
| S +100y | **2.08 ± .01** | 2.09 ± .01 | 2.10 ± .02 | 2.09 ± .01 | 2.10 ± .01 | 2.10 ± .02 | 2.09 ± .01 | 2.11 ± .02 |

## 8 RESULTS

Table 1 shows that, across 3 different increments for the oracle, $BB_{surv}$ outperforms other algorithms when budget is equal to 20 across all 3 real world datasets. This is promising as not only does this show that our method beat other metrics, but we also outperform BatchBALD which suggests that the method we used to deal with the incremental gain helped the models performance. In particular we can also see that as the increment increases, the traditional BatchBALD method and our altered method converge as the original BatchBALD is equivalent theoretically to $BB_{surv}$ when $I = \infty$. The Additional Data section B shows more information in results including different metrics, and budgets. There is a very large amount of randomness here, the Bayesian model

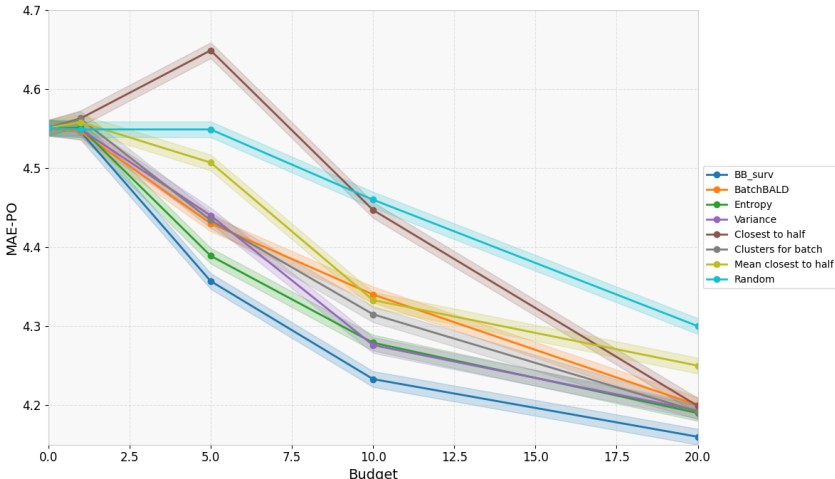

Figure 1: Plot of MAE-PO evaluation of different acquisition functions as a function of budget. Each point is the average of 40 predictions by the model. The plot uses the MIMIC dataset starting with a pool of 900 censored and 100 uncensored points. The increment is 5 years. Each instance costs the same.

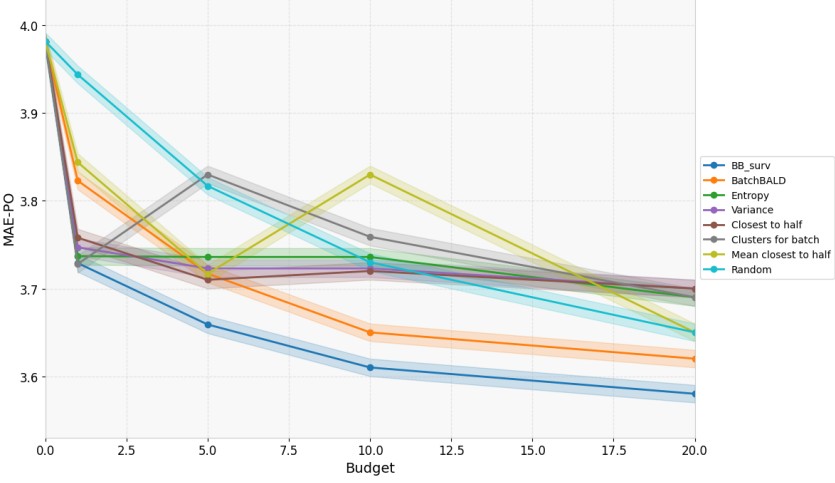

Figure 2: Plot of MAE-PO evaluation of different acquisition functions as a function of budget. Each point is the average of 40 predictions by the model. The plot uses the NACD dataset starting with a pool of 900 censored and 100 uncensored points. The increment is 10 years. Each instance has a random cost between 0.2 and 0.8.

itself provides different predictions and we made each evaluation as an average over 40 predictions. Furthermore, there is inherent randomness as to what instances you initially give to the model. For BatchBALD and in turn our model to succeed, you cannot have so few points that the models predictions are entirely inaccurate, if this is the case we recommend sampling randomly for some time as it may be better (Yehuda et al., 2022). Furthermore the model also cannot have too many data instances to start as then no method works as the model has converged already. We initialized our models with 100 instances uncensored and 900 censored in the training data (enough must be in the pool so that the acquisition functions have more to choose from). One final point here is that in table 1, MCtH does occasionally tie with $BB_{surv}$ for the lowest MAE-PO values. MCtH is a more involved method and in certain budget settings it does seem to do surprisingly well. This is a surprising finding, however this pattern does not hold in the non-uniform costs setting.

In figure 1 we can see that across budgets for all 3 datasets, $BB_{surv}$ does the best, however a further note is that there is more inherent randomness when we compare across budget as each time represents a new training of a model rather than the same model trained further.

In figure 2 we can see that the same results as the uniform costs case seem to present themselves in the non-uniform costs case. Except this time we note that $BB_{surv}$ has an even more distinct advantage. We believe this is because the method of dealing with the budget for $BB_{surv}$ is developed for based off of reducing the problem to the maximum coverage problem. Since many of the other metrics do not measure mutual information, then perhaps they do not reduce to the maximum coverage problem and thus they need to handle the non-uniform costs differently. Mutual information is an incredibly flexible metric that allows for us to easily handle budget in our method.

For MIMIC, a larger budget was needed for both settings due to the complexity of the dataset, thus in all our results although the budget is the same, the cost of instances was one fifth as much as the other cases. These results held consistently for other evaluation metrics including concordance, as well as MAE measured without including the survival test data.

## 9 CONCLUSION

In this paper, we discussed a generalized form of active learning that incorporates budget constraints and, when necessary, accounts for the individual costs of queried instances. We explored methods to extend acquisition functions for use with censored data and to account for scenarios where only partial information is gained during queries. Our proposed method was evaluated across three real-world datasets; however, we anticipate that this emerging area of research will inspire many future studies.

In particular, one promising direction is to combine $BB_{surv}$ with a semi-supervised approach. As noted in Kirsch et al. (2019), such an approach may enhance performance, and given the success of MCtH, we believe this is a worthwhile avenue to explore. Additionally, it would be valuable to revisit some of our assumptions, such as whether the increment changes over time or if data is censored in ways other than randomly.

Finally, there is a substantial body of literature on methods for approximating the maximum coverage problem. Alternative approximation schemes beyond the greedy approach may prove advantageous and warrant further investigation in future work.

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

# A    THEORETICAL ANALYSIS

## A.1    REDUCING BATCH SELECTION TO WEIGHTED MAX COVER

The maximum coverage problem is a well known combinatorial problem that involves choosing k sets, from a group of N sets of integers. The task is to choose the k sets whose union is maximal. For example if the sets are as follows: $S_1 = \{1, 2, 3\}$, $S_2 = \{2, 3, 4\}$, $S_3 = \{4, 5\}$, $S_4 = \{6\}$, where here N=4 and k=2, then the optimal choice of sets here are sets $S_1$ and $S_3$, as their union $\{1, 2, 3, 4, 5\}$ is larger than the union of $S_1$ and $S_2$: $\{1, 2, 3, 4\}$ or $S_2$ and $S_3 = \{2, 3, 4, 5\}$, and all pairs with $S_4$ make at max 4. This problem is known to be np-hard to solve optimally.

There is an extension of this problem called the "weighted" maximum cover problem where everything is the same except each integer has attached to it a weight, and the goal is now to maximize the sum of the weight of the union rather than simply the size of the union. In the example above, if the integers 1,2,3,4, and 5 all had weight 1, but 6 had weight 10, then now we certainly would wish to include $S_4$ as part of one of the sets we choose, in this case you could choose $S_1$ and $S_4$, or $S_2$ and $S_4$, as both would give you a total highest weight of 13.

If you have a set of data points $D$, then each data point $d_i \in D$, provides some amount of information. Any two data points $d_i$ and $d_j$ also have an intersection to consider. Indeed for any batch of data points,we consider information as area that is covered by an area in the information space.

If we now label each unique intersection as an integer, then we now have a bunch of sets, each filled with integers. Furthermore we can give each integer a weight as the amount of area they cover in information space (the amount of information they are expected to give the model). In the batch active learning case, the task now becomes choosing the k data points (which are sets in this case) out of the N total data points that give maximal information cover maximal area). Thus we can simplify the active learning problem using this formulation of mutual information provided by BatchBALD into the weighted maximum coverage problem discussed above.

This representation allows us to see that this problem, at least with the information given, is NP-hard as the maximum coverage probelm is NP-hard. In future works maybe it turns out that most datasets have an underlying structure that makes batch selection easier, however in this general case we can argue NP-hardness This is a very useful representation of the problem as it allows an easier way to think about the machine learning problem as a simpler combinatorial one. Furthermore there exists literature in this field that we can use.

One such insight from literature is that if the information function is a submodular function, then the greedy algorithm does very well.

A set function $f : 2^N \to R$ defined on the subsets of a finite set $N$ is called **submodular** if for every $A, B \subseteq N$,

$$f(A) + f(B) \quad \geq \quad f(A \cup B) + f(A \cap B).$$

Equivalently, $f$ is submodular if it satisfies the **diminishing returns** property: for every $A \subseteq B \subseteq N$ and $x \in N \setminus B$,

$$f(A \cup \{x\}) - f(A) \quad \geq \quad f(B \cup \{x\}) - f(B).$$

The greedy approach (Khuller et al., 1999), gives the highest known guaranteed lower bound of all polynomial time approximation schemes (Khuller et al., 1999). In fact proving there is a polynomial time approximation scheme that achieves a higher lower bound is equivalent to proving P = NP.

The greedy algorithm achieves a lower bound of 1-1/e which is about 63% of the optimal. And indeed is the algorithm used in BatchBALD for selecting it's batch.

Another advantage of this formulation is it allows us extend to the non-uniform case. There is an extension of the weighted maximum cover problem known as the "budgeted" maximum cover problem where along with the constrained from the weighted problem a budget is also given and each set has a given cost attached.

Khuller et al. (1999) have provided a modified greedy algorithm to the budgeted problem that also meets the lower bound of 1-1/e. They further showed that finding a polynomial time algorithm that achieves a better lower bound in this setting would be equivalent to proving $NP \subseteq \text{Dtime}(n^{\log \log n})$.

## A.2 Changing the Budgeted Maximum Coverage Algorithm

In Khuller et al. (1999), a novel greedy algorithm for the budgeted maximum cover case is provided that still provides the same lower bound guarantees as the original greedy algorithm. We illustrate this novel algorithm in Algorithm 2.

---

**Algorithm 2** Optimal 1 - 1/e-approximate algorithm for Budgeted Maximal Coverage

---

**Require:** Pool of points $S$, budget $B$, weights $w_i$, costs $c_i$, subset size $k$
1: $H_1 \leftarrow \arg\max\{w(G) : G \subseteq S, |G| < k, c(G) \leq B\}$
2: $H_2 \leftarrow \emptyset$
3: **for all** $G \subseteq S$ such that $|G| = k$ and $c(G) \leq B$ **do**
4:    $U \leftarrow S \setminus G$
5:    **repeat**
6:       Select $x_i \in U$ that maximizes $\frac{w_i'}{c_i}$
7:       **if** $c(G) + c_i \leq B$ **then**
8:          $G \leftarrow G \cup x_i$
9:          $U \leftarrow U \setminus x_i$
10:      **end if**
11:    **until** $U = \emptyset$ or $c(G) + c_i > B$
12:    **if** $w(G) > w(H_2)$ **then**
13:       $H_2 \leftarrow G$
14:    **end if**
15: **end for**
16: **if** $w(H_1) > w(H_2)$ **then**
17:    **Output:** $H_1$
18: **else**
19:    **Output:** $H_2$
20: **end if**

---

*m:*In our own notation, the pool of points $S = D_{pool}$, and the weights $w_i$ is estimated using our acquisition function $a_{BB_{surv}}$. k is meant to be a parameter chosen by the user where a higher k yields better performance however at a higher computational cost. We can take k=3 which is the lowest k that guarantees the optimal $1 - 1/e$ bound. This new greedy algorithm is very computationally expensive. The first part of the algorithm relies on finding the set out of all sets of size less than $k = 3$ size (via brute force) that maximizes the weight, and assigning it to $H1$ which has a complexity of $O(n^2)$ if we denote $n = |D_{pool}|$. The rest of the algorithm involves for every possible initial set of size $k = 3$ instances, greedily adding instances to this selected batch based off of the ratio of the weight to cost of an instance, which is of order $O(n^3)$. We argue that for the purposes of our settings, we do not need the first step and we can greatly reduce the complexity of the second step. The arguments we are making are not guaranteed or proven, however it has shown results experimentally and serves as a strong approximation of this algorithm in most settings.

For the first part of the algorithm, in deep learning models, often two or 3 points are not significant enough to significantly alter the models loss Domingos (2012). The first part of the algorithm is only relevant if it turns out that a set of size 3 points gives more information than a set of larger size. In deep learning models all data instances provide some information, and when accounting for the weight to cost ratio, it is very likely that the points within $H1$ are also selected in sets of larger size. Thus we omit this step in the algorithm, especially with larger budgets it is very unlikely to help.

The second part of the algorithm which involves for every possible initial set of size $k = 3$ instances, greedily adding instances to this selected batch based off of the ratio of the weight to cost of an instance. The main part that dramatically increases the computational complexity here is having to consider all $n^3$ starting positions. In Khuller et al. (1999) they discuss one main setting where considering all possible starting states is useful, is if the points you are choosing from are dense in information space, that is that the sets share a very high amount of similar elements between them and learning about any one instance implies you learn about the larger group. In our setting, we start out with a small number of instances and wish to learn about a complex function that covers a large amount in information space, the instances in the pool we can learn from are often very sparse and give us vastly different information from different parts of the information space rather than all being

dense and giving us the same types of information. Under these settings you often are not choosing multiple instances from a clump of points but rather want to spread out your queried instances. Thus considering all possible starting points is not efficient and we recommend omiting it all together.

If we omit these two elements from Algorithm 2, we end up with the method described in the paper shown in Algorithm 1.

### A.3 Choosing the Same Instance Multiple Times in One Query

Since in our setting the Oracle does not give full information. It opens up the question as to if you can choose the same instance multiple times in one query. For example, if the Oracle gives 5 years of new information for Alice, you might want to ask the Oracle to give you 10 years but for double the cost of course. This is not often seen in the real world, perhaps if you are working with a software that has all the instances but only gives incremental updates. Or maybe when dealing with "label delays", covered in Dedja et al. (2023).

Fortunately, our formulation allows for generalization over this as well. Instead of considering how many times we should ask for the "Alice" instance, we can see it such that asking for two increments of Alice, is a seperate instance with its own cost as asking for one increment of Alice. Furthermore, the entire information that can be gained from for the one increment instance is contained in asking for the 2 increment instance. The question now becomes is the extra information work the cost? Since the costs between these two instances is different, we again have a case of the budgeted maximum coverage problem, for which we show in the paper that there already exists an optimal approximation greedy algorithm for this setting. A final note is that there is never an incentive to ask for both the Alice with 2 increments instance and the Alice with 1 increment instance, but the optimal greedy algorithm does not assume this and may take this action. Thus we can actually create an even better algorithm than greedy in this specific case by only allowing it to ever choose the maximum number of increments instance each time.

## B    Additional Data

We show here some additional results for the two settings. We have only included results we find most interesting, however we will submit all our results and code upon acceptance.

Table 2: MAE-PO across datasets and increments, budget = 0. Uniform setting.

| Dataset | BB surv | BatchBALD | Entropy | Var | CtH | CfB | MCtH | Random |
|---|---|---|---|---|---|---|---|---|
| M +5 years | 4.55 ± 0.05 | 4.55 ± 0.05 | 4.55 ± 0.05 | 4.55 ± 0.05 | 4.55 ± 0.05 | 4.55 ± 0.05 | 4.55 ± 0.05 | 4.55 ± 0.05 |
| M +10 years | 4.55 ± 0.05 | 4.55 ± 0.05 | 4.55 ± 0.05 | 4.55 ± 0.05 | 4.55 ± 0.05 | 4.55 ± 0.05 | 4.55 ± 0.05 | 4.55 ± 0.05 |
| M +100 years | 4.55 ± 0.05 | 4.55 ± 0.05 | 4.55 ± 0.05 | 4.55 ± 0.05 | 4.55 ± 0.05 | 4.55 ± 0.05 | 4.55 ± 0.05 | 4.55 ± 0.05 |
| N +5 years | 3.98 ± 0.03 | 3.98 ± 0.03 | 3.98 ± 0.03 | 3.98 ± 0.03 | 3.98 ± 0.03 | 3.98 ± 0.03 | 3.98 ± 0.03 | 3.98 ± 0.03 |
| N +10 years | 3.98 ± 0.03 | 3.98 ± 0.03 | 3.98 ± 0.03 | 3.98 ± 0.03 | 3.98 ± 0.03 | 3.98 ± 0.03 | 3.98 ± 0.03 | 3.98 ± 0.03 |
| N +100 years | 3.98 ± 0.03 | 3.98 ± 0.03 | 3.98 ± 0.03 | 3.98 ± 0.03 | 3.98 ± 0.03 | 3.98 ± 0.03 | 3.98 ± 0.03 | 3.98 ± 0.03 |
| S +5 years | 2.12 ± 0.02 | 2.12 ± 0.02 | 2.12 ± 0.02 | 2.12 ± 0.02 | 2.12 ± 0.02 | 2.12 ± 0.02 | 2.12 ± 0.02 | 2.12 ± 0.02 |
| S +10 years | 2.12 ± 0.02 | 2.12 ± 0.02 | 2.12 ± 0.02 | 2.12 ± 0.02 | 2.12 ± 0.02 | 2.12 ± 0.02 | 2.12 ± 0.02 | 2.12 ± 0.02 |
| S +100 years | 2.12 ± 0.02 | 2.12 ± 0.02 | 2.12 ± 0.02 | 2.12 ± 0.02 | 2.12 ± 0.02 | 2.12 ± 0.02 | 2.12 ± 0.02 | 2.12 ± 0.02 |

Table 3: MAE-PO across datasets and increments, budget = 5. Uniform setting.

| Dataset | BB surv | BatchBALD | Entropy | Variance | CtH | CfB | MCtH | Random |
|---|---|---|---|---|---|---|---|---|
| M +5 years | 4.36 ± 0.03 | 4.43 ± 0.03 | 4.39 ± 0.03 | 4.44 ± 0.03 | 4.65 ± 0.04 | 4.43 ± 0.03 | 4.51 ± 0.03 | 4.51 ± 0.03 |
| M +10 years | 4.37 ± 0.03 | 4.42 ± 0.03 | 4.31 ± 0.03 | 4.39 ± 0.03 | 4.37 ± 0.03 | 4.37 ± 0.03 | 4.42 ± 0.03 | 4.42 ± 0.03 |
| M +100 years | 4.21 ± 0.04 | 4.26 ± 0.04 | 4.26 ± 0.04 | 4.37 ± 0.03 | 4.47 ± 0.03 | 4.29 ± 0.03 | 4.25 ± 0.03 | 4.25 ± 0.03 |
| N +5 years | 3.82 ± 0.03 | 3.83 ± 0.03 | 3.83 ± 0.03 | 3.85 ± 0.03 | 3.85 ± 0.03 | 3.94 ± 0.03 | 3.95 ± 0.03 | 3.95 ± 0.03 |
| N +10 years | 3.74 ± 0.03 | 3.73 ± 0.03 | 3.76 ± 0.03 | 3.79 ± 0.03 | 3.85 ± 0.03 | 3.90 ± 0.03 | 3.75 ± 0.03 | 3.75 ± 0.03 |
| N +100 years | 3.78 ± 0.03 | 3.78 ± 0.03 | 3.79 ± 0.03 | 3.88 ± 0.03 | 3.88 ± 0.03 | 3.81 ± 0.03 | 3.79 ± 0.03 | 3.79 ± 0.03 |
| S +5 years | 2.09 ± 0.01 | 2.11 ± 0.01 | 2.11 ± 0.02 | 2.11 ± 0.02 | 2.11 ± 0.02 | 2.11 ± 0.02 | 2.10 ± 0.01 | 2.10 ± 0.02 |
| S +10 years | 2.09 ± 0.01 | 2.10 ± 0.02 | 2.11 ± 0.01 | 2.10 ± 0.02 | 2.10 ± 0.02 | 2.11 ± 0.01 | 2.10 ± 0.02 | 2.10 ± 0.02 |
| S +100 years | 2.08 ± 0.01 | 2.10 ± 0.02 | 2.11 ± 0.02 | 2.10 ± 0.02 | 2.10 ± 0.01 | 2.11 ± 0.02 | 2.10 ± 0.01 | 2.10 ± 0.02 |

Table 4: MAE-PO across datasets and increments, budget = 10. Uniform setting.

| Dataset | BB surv | BatchBALD | Entropy | Var | CtH | CfB | MCtH | Random |
|---|---|---|---|---|---|---|---|---|
| M +5y | 4.23 ± .01 | 4.34 ± .02 | 4.28 ± .01 | 4.28 ± .02 | 4.45 ± .01 | 4.32 ± .02 | 4.23 ± .02 | 4.46 ± .01 |
| M +10y | 4.26 ± .01 | 4.35 ± .02 | 4.28 ± .01 | 4.33 ± .02 | 4.31 ± .01 | 4.27 ± .02 | 4.28 ± .01 | 4.28 ± .02 |
| M +100y | 4.18 ± .02 | 4.18 ± .01 | 4.18 ± .02 | 4.27 ± .01 | 4.27 ± .02 | 4.23 ± .01 | 4.18 ± .02 | 4.18 ± .01 |
| N +5y | 3.63 ± .01 | 3.63 ± .02 | 3.64 ± .01 | 3.66 ± .02 | 3.69 ± .01 | 3.70 ± .02 | 3.81 ± .01 | 3.89 ± .02 |
| N +10y | 3.59 ± .01 | 3.60 ± .02 | 3.61 ± .01 | 3.65 ± .02 | 3.71 ± .01 | 3.77 ± .02 | 3.58 ± .01 | 3.74 ± .02 |
| N +100y | 3.67 ± .01 | 3.66 ± .01 | 3.68 ± .01 | 3.73 ± .02 | 3.73 ± .01 | 3.65 ± .02 | 3.68 ± .01 | 3.70 ± .02 |
| S +5y | 2.09 ± .01 | 2.11 ± .01 | 2.12 ± .01 | 2.10 ± .01 | 2.10 ± .01 | 2.10 ± .01 | 2.11 ± .01 | 2.12 ± .01 |
| S +10y | 2.09 ± .01 | 2.09 ± .02 | 2.10 ± .01 | 2.09 ± .01 | 2.09 ± .02 | 2.10 ± .01 | 2.09 ± .01 | 2.11 ± .02 |
| S +100y | 2.08 ± .01 | 2.09 ± .01 | 2.10 ± .02 | 2.09 ± .01 | 2.10 ± .01 | 2.10 ± .02 | 2.09 ± .01 | 2.11 ± .02 |

Table 5: MAE-PO across datasets and increments, budget = 500 (to show convergence). Uniform setting.

| Dataset | BB surv | BatchBALD | Entropy | Variance | CtH | CfB | MCtH | Random |
|---|---|---|---|---|---|---|---|---|
| M +5 years | 3.89 ± 0.02 | 3.89 ± 0.02 | 3.89 ± 0.02 | 3.89 ± 0.02 | 3.89 ± 0.02 | 3.89 ± 0.02 | 3.89 ± 0.02 | 3.89 ± 0.02 |
| M +10 years | 3.89 ± 0.02 | 3.89 ± 0.02 | 3.89 ± 0.02 | 3.89 ± 0.02 | 3.88 ± 0.02 | 3.89 ± 0.02 | 3.89 ± 0.02 | 3.88 ± 0.02 |
| M +100 years | 3.85 ± 0.02 | 3.85 ± 0.02 | 3.85 ± 0.02 | 3.85 ± 0.02 | 3.85 ± 0.02 | 3.85 ± 0.02 | 3.85 ± 0.02 | 3.85 ± 0.02 |
| N +5 years | 3.30 ± 0.02 | 3.30 ± 0.02 | 3.30 ± 0.02 | 3.30 ± 0.02 | 3.31 ± 0.02 | 3.30 ± 0.02 | 3.30 ± 0.02 | 3.30 ± 0.02 |
| N +10 years | 3.27 ± 0.02 | 3.27 ± 0.02 | 3.27 ± 0.02 | 3.27 ± 0.02 | 3.28 ± 0.02 | 3.27 ± 0.02 | 3.27 ± 0.02 | 3.28 ± 0.02 |
| N +100 years | 3.33 ± 0.02 | 3.33 ± 0.02 | 3.33 ± 0.02 | 3.33 ± 0.02 | 3.34 ± 0.02 | 3.33 ± 0.02 | 3.33 ± 0.02 | 3.34 ± 0.02 |
| S +5 years | 1.88 ± 0.02 | 1.88 ± 0.02 | 1.88 ± 0.02 | 1.78 ± 0.01 | 1.88 ± 0.01 | 1.78 ± 0.01 | 1.88 ± 0.02 | 1.88 ± 0.01 |
| S +10 years | 1.78 ± 0.01 | 1.88 ± 0.01 | 1.81 ± 0.01 | 1.78 ± 0.01 | 1.88 ± 0.01 | 1.88 ± 0.01 | 1.88 ± 0.01 | 1.81 ± 0.01 |
| S +100 years | 1.78 ± 0.02 | 1.78 ± 0.01 | 1.78 ± 0.02 | 1.79 ± 0.01 | 1.79 ± 0.02 | 1.78 ± 0.01 | 1.89 ± 0.02 | 1.79 ± 0.02 |

When evaluated on other forms of MAE (without including survival data in the dataset) we saw very similar results as the ones presented in this section. However we felt it important to also include a measure of concordance. We have tables of MAE-PO for various budgets on both uniform and non-uniform settings here.

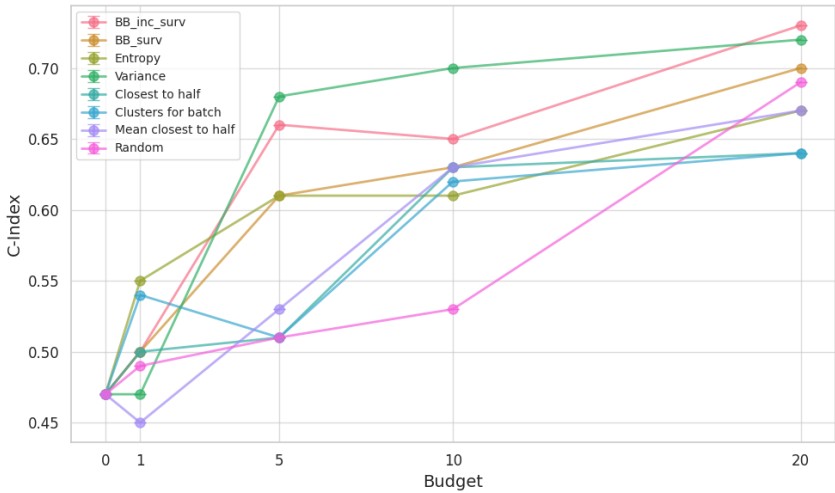

Figure 3: Plot of MAE-PO evaluation of different acquisition functions as a function of budget. Each point is the average of 40 predictions by the model. The plot uses the NACD dataset starting with a pool of 900 censored and 100 uncensored points. The increment is 10 years. Each instance has a random cost between 0.2 and 0.8

Table 6: MAE-PO across datasets and increments, budget = 0. Non-Uniform setting.

| Dataset | BB surv | BatchBALD | Entropy | Variance | CtH | CfB | MCtH |
|---|---|---|---|---|---|---|---|
| M +5 years | 4.55 ± 0.05 | 4.55 ± 0.05 | 4.55 ± 0.05 | 4.55 ± 0.05 | 4.55 ± 0.05 | 4.55 ± 0.05 | 4.55 ± 0.05 |
| M +10 years | 4.55 ± 0.05 | 4.55 ± 0.05 | 4.55 ± 0.05 | 4.55 ± 0.05 | 4.55 ± 0.05 | 4.55 ± 0.05 | 4.55 ± 0.05 |
| M +100 years | 4.55 ± 0.05 | 4.55 ± 0.05 | 4.55 ± 0.05 | 4.55 ± 0.05 | 4.55 ± 0.05 | 4.55 ± 0.05 | 4.55 ± 0.05 |
| N +5 years | 3.98 ± 0.03 | 3.98 ± 0.03 | 3.98 ± 0.03 | 3.98 ± 0.03 | 3.98 ± 0.03 | 3.98 ± 0.03 | 3.98 ± 0.03 |
| N +10 years | 3.98 ± 0.03 | 3.98 ± 0.03 | 3.98 ± 0.03 | 3.98 ± 0.03 | 3.98 ± 0.03 | 3.98 ± 0.03 | 3.98 ± 0.03 |
| N +100 years | 3.98 ± 0.03 | 3.98 ± 0.03 | 3.98 ± 0.03 | 3.98 ± 0.03 | 3.98 ± 0.03 | 3.98 ± 0.03 | 3.98 ± 0.03 |
| S +5 years | 2.11 ± 0.02 | 2.11 ± 0.02 | 2.11 ± 0.02 | 2.11 ± 0.02 | 2.11 ± 0.02 | 2.11 ± 0.02 | 2.11 ± 0.02 |
| S +10 years | 2.11 ± 0.02 | 2.11 ± 0.02 | 2.11 ± 0.02 | 2.11 ± 0.02 | 2.11 ± 0.02 | 2.11 ± 0.02 | 2.11 ± 0.02 |
| S +100 years | 2.11 ± 0.02 | 2.11 ± 0.02 | 2.11 ± 0.02 | 2.11 ± 0.02 | 2.11 ± 0.02 | 2.11 ± 0.02 | 2.11 ± 0.02 |

Table 7: MAE-PO across datasets and increments, budget = 5. Non-Uniform setting.

| Dataset | BB surv | BatchBALD | Entropy | Variance | CtH | CfB | MCtH | Random |
|---|---|---|---|---|---|---|---|---|
| M +5 years | 4.21 ± 0.05 | 4.23 ± 0.05 | 4.51 ± 0.05 | 4.35 ± 0.05 | 4.39 ± 0.05 | 4.29 ± 0.05 | 4.58 ± 0.05 | 4.59 ± 0.05 |
| M +10 years | 4.32 ± 0.05 | 4.35 ± 0.05 | 4.59 ± 0.05 | 4.61 ± 0.05 | 4.59 ± 0.05 | 4.59 ± 0.05 | 4.62 ± 0.05 | 4.62 ± 0.05 |
| M +100 years | 4.32 ± 0.05 | 4.33 ± 0.05 | 4.34 ± 0.05 | 4.35 ± 0.05 | 4.45 ± 0.05 | 4.33 ± 0.05 | 4.39 ± 0.05 | 4.43 ± 0.05 |
| N +5 years | 3.88 ± 0.03 | 3.92 ± 0.03 | 3.91 ± 0.03 | 3.87 ± 0.03 | 3.88 ± 0.03 | 3.88 ± 0.03 | 3.95 ± 0.03 | 3.97 ± 0.03 |
| N +10 years | 3.73 ± 0.03 | 3.82 ± 0.03 | 3.74 ± 0.03 | 3.75 ± 0.03 | 3.76 ± 0.03 | 3.73 ± 0.03 | 3.84 ± 0.03 | 3.84 ± 0.03 |
| N +100 years | 3.73 ± 0.03 | 3.73 ± 0.03 | 3.74 ± 0.03 | 3.75 ± 0.03 | 3.76 ± 0.03 | 3.73 ± 0.03 | 3.84 ± 0.03 | 3.84 ± 0.03 |
| S +5 years | 2.09 ± 0.02 | 2.10 ± 0.02 | 2.10 ± 0.02 | 2.09 ± 0.02 | 2.09 ± 0.02 | 2.11 ± 0.02 | 2.10 ± 0.02 | 2.11 ± 0.02 |
| S +10 years | 2.09 ± 0.02 | 2.10 ± 0.02 | 2.10 ± 0.02 | 2.11 ± 0.02 | 2.09 ± 0.02 | 2.11 ± 0.02 | 2.10 ± 0.02 | 2.11 ± 0.02 |
| S +100 years | 2.09 ± 0.02 | 2.11 ± 0.02 | 2.10 ± 0.02 | 2.10 ± 0.02 | 2.09 ± 0.02 | 2.11 ± 0.02 | 2.10 ± 0.02 | 2.11 ± 0.02 |

Table 8: MAE-PO across datasets and increments, budget = 10. Non-Uniform setting.

| Dataset | BB surv | BatchBALD | Entropy | Variance | CtH | CfB | MCtH | Random |
|---|---|---|---|---|---|---|---|---|
| M +5 years | 4.17 ± 0.05 | 4.18 ± 0.05 | 4.25 ± 0.05 | 4.35 ± 0.05 | 4.37 ± 0.05 | 4.34 ± 0.05 | 4.39 ± 0.05 | 4.49 ± 0.05 |
| M +10 years | 3.97 ± 0.05 | 4.00 ± 0.05 | 3.97 ± 0.05 | 4.10 ± 0.05 | 4.10 ± 0.05 | 3.96 ± 0.05 | 4.15 ± 0.05 | 4.25 ± 0.05 |
| M +100 years | 4.00 ± 0.05 | 4.01 ± 0.05 | 4.05 ± 0.05 | 4.14 ± 0.05 | 4.19 ± 0.05 | 4.03 ± 0.05 | 4.07 ± 0.05 | 4.07 ± 0.05 |
| N +5 years | 3.81 ± 0.03 | 3.83 ± 0.03 | 3.84 ± 0.03 | 3.81 ± 0.03 | 3.81 ± 0.03 | 3.83 ± 0.03 | 3.73 ± 0.03 | 3.83 ± 0.03 |
| N +10 years | 3.72 ± 0.03 | 3.72 ± 0.03 | 3.74 ± 0.03 | 3.72 ± 0.03 | 3.72 ± 0.03 | 3.76 ± 0.03 | 3.83 ± 0.03 | 3.76 ± 0.03 |
| N +100 years | 3.71 ± 0.02 | 3.74 ± 0.02 | 3.73 ± 0.02 | 3.80 ± 0.02 | 3.73 ± 0.02 | 3.66 ± 0.02 | 3.77 ± 0.02 | 3.66 ± 0.02 |
| S +5 years | 2.08 ± 0.02 | 2.10 ± 0.02 | 2.10 ± 0.02 | 2.10 ± 0.02 | 2.10 ± 0.02 | 2.10 ± 0.02 | 2.10 ± 0.02 | 2.12 ± 0.02 |
| S +10 years | 2.09 ± 0.02 | 2.10 ± 0.02 | 2.10 ± 0.02 | 2.09 ± 0.02 | 2.10 ± 0.02 | 2.10 ± 0.02 | 2.10 ± 0.02 | 2.12 ± 0.02 |
| S +100 years | 2.08 ± 0.02 | 2.10 ± 0.02 | 2.10 ± 0.02 | 2.10 ± 0.02 | 2.10 ± 0.02 | 2.10 ± 0.02 | 2.10 ± 0.02 | 2.10 ± 0.02 |

Table 9: MAE-PO across datasets and increments, budget = 500 (for convergence). Non-Uniform setting.

| Dataset | BB surv | BatchBALD | Entropy | Variance | CtH | CfB | MCtH | Random |
|---|---|---|---|---|---|---|---|---|
| M +5 years | 4.19 ± 0.02 | 4.19 ± 0.02 | 4.19 ± 0.02 | 4.19 ± 0.02 | 4.19 ± 0.02 | 4.19 ± 0.02 | 4.19 ± 0.02 | 4.19 ± 0.02 |
| M +10 years | 4.23 ± 0.02 | 4.23 ± 0.02 | 4.23 ± 0.02 | 4.23 ± 0.02 | 4.23 ± 0.02 | 4.23 ± 0.02 | 4.23 ± 0.02 | 4.23 ± 0.02 |
| M +100 years | 4.15 ± 0.02 | 4.15 ± 0.02 | 4.15 ± 0.02 | 4.15 ± 0.02 | 4.15 ± 0.02 | 4.15 ± 0.02 | 4.15 ± 0.02 | 4.15 ± 0.02 |
| N +5 years | 3.60 ± 0.02 | 3.60 ± 0.02 | 3.60 ± 0.02 | 3.60 ± 0.02 | 3.61 ± 0.02 | 3.60 ± 0.02 | 3.60 ± 0.02 | 3.60 ± 0.02 |
| N +10 years | 3.57 ± 0.02 | 3.57 ± 0.02 | 3.57 ± 0.02 | 3.57 ± 0.02 | 3.58 ± 0.02 | 3.57 ± 0.02 | 3.57 ± 0.02 | 3.57 ± 0.02 |
| N +100 years | 3.63 ± 0.02 | 3.63 ± 0.02 | 3.63 ± 0.02 | 3.63 ± 0.02 | 3.64 ± 0.02 | 3.63 ± 0.02 | 3.63 ± 0.02 | 3.63 ± 0.02 |
| S +5 years | 2.08 ± 0.01 | 2.08 ± 0.01 | 2.08 ± 0.01 | 2.08 ± 0.01 | 2.08 ± 0.01 | 2.08 ± 0.01 | 2.08 ± 0.01 | 2.08 ± 0.01 |
| S +10 years | 2.08 ± 0.01 | 2.08 ± 0.01 | 2.08 ± 0.01 | 2.08 ± 0.01 | 2.08 ± 0.01 | 2.08 ± 0.01 | 2.08 ± 0.01 | 2.08 ± 0.01 |
| S +100 years | 2.09 ± 0.01 | 2.08 ± 0.01 | 2.08 ± 0.01 | 2.09 ± 0.01 | 2.09 ± 0.01 | 2.08 ± 0.01 | 2.09 ± 0.01 | 2.09 ± 0.01 |

