# OpenReview forum: "Budget-constrained Active Learning to De-censor Survival Data"
_ICLR.cc/2025/Conference — Submitted to ICLR 2025_

### Official Review · Reviewer_ViVr · 2024-10-29

**Soundness:** 1
**Presentation:** 2
**Contribution:** 2
**Rating:** 3
**Confidence:** 4

**Summary:**

This paper proposes an active learning (AL) approach for survival analysis data, where the AL is constrained to a budget. The proposed approach is essentially an extension/adaptation of the BatchBALD algorithm (Kirsch et al., 2019) to account for survival data and a limited budget. The main motivation and application comes from the medical domain, where the proposed approach aims to identify which patients to follow up with to maximize the mutual information criterium of BatchBALD. The empirical results based on 3 real-world datasets in the medical domain show that the proposed approach can outperform standard BatchBALD, as well as 5 additional naive baselines.

**Strengths:**

1) The **problem addressed in this paper is relevant** and the existing research on the topic of AL for survival data is indeed limited.
2) The proposed approach is **an original combination of existing ideas**, namely budgeted learning and BatchBALD for AL, with the latter being "tweaked" to work with survival data.

**Weaknesses:**

1) The **contributions are not very clear**. The authors should clearly state which contributions they are claiming. Is the extension of BatchBALD to a budget-constrained setting the central contribution? If so, why is the theoretical analysis included in the appendix rather than in the main text? Is the adaptation of BatchBALD to survival data the central contribution? Both? The narration is ambiguous with respect to this, thus unnecessarily increasing the difficulty of assessing novelty.
2) The **novelty is unfortunately low**. It is not clear how the proposed approach for accounting for the budget constraint is different from the existing approaches for budgeted learning from the literature. The adaptation of BatchBALD proposed for survival data is also very incremental and not adequately grounded. The authors also mention that "so far no method has been developed looking at
budgeted learning with survival data" - however, there are other works in the literature looking at applying BALD for right-censored data (see e.g., https://arxiv.org/pdf/2402.11973). How does the proposed approach relate to these works?
3) There are **several concerns regarding the soundness** of the proposed approach. The extension of BatchBALD to a budget-constrained setting is not clearly described and is not adequately grounded. The proposed approach is an adaptation of the algorithm proposed in (Khuller et al., 1999) "by omitting the first two lines". However, it is not clear what lines the authors refer to and, most importantly, what are the theoretical grounds for doing so. Similarly, the adaptions of BatchBALD for survival data are essentially presented as "algorithmic tweaks" (e.g., setting probabilities to zero and renormalizing) without a proper theoretical justification. Lastly, it is not clear how the authors handle the fact that the queried instances can still be censored - are the censored times assumed to be known a priori before querying? If so, how limiting is that assumption in terms of the applicability of the proposed approach?
4) The **limited application scope** of the proposed approach is also a topic of concern. Although I agree with the authors that AL for survival data is a topic worth investigating, and the literature is indeed lacking on this, the application scope is, in my opinion, broader than the medical domain application presented in the paper. This limits the target audience of the paper quite a lot. I would encourage the authors to consider revising their presentation to broaden the scope of application to other domains as well. For example, the authors mention applications in finance and engineering in the abstract - it would be interesting to discuss those as well. Lastly, the authors consider only approaches where time is discretized into bins (they use Multi-Task Logistic Regression as the underlying survival model), and the proposed adaptation of BatchBALD to handle survival data in Section 5 relies on this assumption, which can be quite limiting. This should be discussed in the paper.
5) The presented **claims are not adequately supported by empirical evidence**. Looking at the results in Table 1, the improvements over standard BatchBALD are very marginal. However, the text claims that "BBsurv outperforms other algorithms when budget is equal to 20 across all 3 real world datasets", which does not seem to always be the case. In fact, even "Entropy" seems to perform quite comparably - did the authors perform a statistical significance test? Confusingly, the Table 1 caption states that "Budget = 10", which is inconsistent with the text. Moreover, the results in Table 1 seem to be inconsistent with Figures 1 and 2. If one considered a vertical line at Budget = 10 (or 20, whichever it is), shouldn't the values and relative order of the approaches be consistent with Table 1?
6) The **presentation can be significantly improved**. There are several typos, poorly constructed sentences, incorrect grammar usage, sentences that are too informal for scientific paper, etc. I strongly encourage the authors to carefully revise the writing and overall presentation of the paper. Other aspects of the presentation, such as mathematical derivations, can be significantly improved. For example, the authors should include numbers for the equations. Similarly, it is unclear how the authors transition from the eq. of the mutual information at the beginning of Section 4.1 to the computation of its right-most term 2 equations below. The right term is an expected entropy, while 2 equations below only the entropy is considered. Can you clarify?

**Questions:**

1) Can you please clearly list the main claimed contributions of the paper?
2) There are other works in the literature looking at applying BALD for right-censored data (see e.g., https://arxiv.org/pdf/2402.11973). How does the proposed approach relate to these works?
3) Please clarify the theoretical ground for "omitting the first two lines" of the algorithm of (Khuller et al., 1999) and for setting the probabilities to zero and renormalizing in the adaptation to BatchBALD to survival data?
4) Are the censored times assumed to be known a priori before querying? If so, how limiting is that assumption in terms of the applicability of the proposed approach?
5) How can the proposed approach be generalized beyond discretized time bins?
6) The results in Table 1 seem to be inconsistent with Figures 1 and 2. If one considered a vertical line at Budget = 10 (or 20, whichever it is), shouldn't the values and relative order of the approaches be consistent with Table 1?
7) It is unclear how the authors transition from the eq. of the mutual information at the beginning of Section 4.1 to the computation of its right-most term 2 equations below. The right term is an expected entropy, while 2 equations below only the entropy is considered. Can you clarify?

---

> ### Author Response · Authors · 2024-11-23
> **Response to reviewer**
>
> Thank you for your very thoughtful review.
>
> To address you key points:
>
> 1. The revised Introduction now explicitly mentions our contributions.
>
> 2. That Introduction also includes a section describing the scope of our work. We hope these additions better highlight the novelty of the work.
>
> 3. We have significantly expanded on our approach to generating a new greedy algorithm, as you suggested, in hopes of demonstrating the soundness of the method.
>
> 4. (Refer to point 2.)
>
> 5. This is an excellent concern. Thank you for pointing this out. We are currently investigating it and will get back to you soon.
>
> 6. We have made significant changes in the new draft to improve the paper’s presentation. Thank you for highlighting this.
>
> Your thoughtful review has been immensely helpful in improving the paper, and we sincerely thank you for it.

---

> > ### Comment · Reviewer_ViVr · 2024-11-25
> >
> > I thank the authors for their responses. Although I agree that this is an interesting research topic, the answers provided are not sufficient to address my previous concerns and potentially revise my original score (for example, the authors mention several times the revised version of their paper, but I do not have access to it). However, I look forward to seeing a revised version of this work.

---

### Official Review · Reviewer_3W4U · 2024-10-29

**Soundness:** 2
**Presentation:** 3
**Contribution:** 4
**Rating:** 3
**Confidence:** 2

**Summary:**

This paper focuses on a problem of great practical interest, when the dataset is partially labeled and the data is censored. It proposes a budgeted learning (a subfield of active learning) method in the context of survival analysis.
It's a niche subject, but the method is very new and original, I am convinced it can be used for practical purposes.

I am not familiar with either active learning or survival analysis (so I'll put 2 on the confidence score).

**Strengths:**

- This paper is clear, domains are well introduced (whereas the paper is at the intersection of survival analysis and active learning)
- The results are convincing, and the authors have made the effort to compare themselves with “naive” methods (sanity checks part), even though there is no work dealing with this case.

**Weaknesses:**

- I have some minor comments, it's more a question of presentation than content.

**Questions:**

1. Assumptions:
- for clarity, I think it would be better to present the assumptions in a formal manner: Assumption 1: (...), Assumption 2: (...)
- for the assumption  that censoring is independent of the features, it is common in survival analysis but can they the authors discuss this more in details what it implies in practice by taking an example ?
2. The measure MAE-PO should be described in the main text.
3. Line 388, the authors describe clearly how the real dataset is "censored". However, I think it's a pity that the authors don't discuss enough the realistic aspect of having the selected datasets censored (and partially labeled), perhaps a realistic situation for a dataset could be put forward.

Minor comments:
- line 139: **, where** L is the size of the training data.
- line 165: 3.1 INITIAL ASSUMPTIONS -> there is no 3.2 section, maybe the authors could use \paragraph instead of \subsection
- it may be personal but I don't like the notations $ctime_i$ and $cevent_i$, maybe the authors can use a shorter version like $c^t_i$ and $c^e_i$ (it is just a proposition)

---

> ### Author Response · Authors · 2024-11-23
> **Response to reviewer**
>
> Thank you for your superb review.
>
> The revised version presents our assumptions more formally, provides a clearer discussion on what "censoring independent of features" means (and provides a realistic example), describes MAE-PO earlier in the text, and addressed the two points raised in the minor comments.
>
> Your feedback was highly constructive, and we sincerely thank you for it. Please let us know if there are any other points of confusion in the paper that we can clarify.

---

> ### Comment · Reviewer_3W4U · 2024-11-25
>
> Thank you for your answer.
> As I said, I'm not really familiar with budgeted learning or survival analysis.
> I've read the other comments, and I'm going to reduce my rating to 3, for three main reasons:
> - the assumptions are not sufficiently discussed by the authors. The PdWR reviewer also mentions that the real time requirement in the training data is very restrictive.
> - previous work is not sufficiently discussed. In addition, the ViVr reviewer mentions that the algorithm is merely an adaptation of the algorithm by Khuller et al. (1999).
> - the authors answer quite briefly to all the reviewers' comments, referring only to their new version of the paper. However, we do not yet have access to it.
>
> However, I am convinced that the subject of the paper is very interesting.

---

> ### Author Response · Authors · 2024-11-25
> **Response to Reviewer**
>
> Thank you for your reply. We understand your decision and your reasons. As many reviewer comments on our paper focussed on the notation, word choice, and presentation style, much of our rebuttal depends on the new draft. That new draft defined our assumptions more formally, and includes more discussion on previous works. Note, however, that we have not yet found any work in our setting; while other reviewers recommended a paper similar to what we suggest, the new revision notes that even that paper differs from ours in key ways.
>
> Unfortunately, we presented the requirement for real time poorly to the PdWR reviewer. Due to our notation in the formulation, this description mistakenly suggested that the learner would need to have complete uncensored information about each training instance. However, the core part of our work is showing that this is not necessary – the active learner even can produce effective models with only the standard survival dataset, with the covariates for each instance, and a label for each censored instance indicating when it was censored. It is the Oracle that needs to be able to produce that “true survival time”, when asked.. To demonstrate this approach, we gave our simulator this complete “survival time” information, which it could dispense to the learner, in response to each learner query. But critically, this was the only auxiliary data the active learner had access – n.b., it did not see the complete data.
>
>
> Also, the ViVr reviewer correctly notes that some of our work is related to Khuller et al. (1999). However,  our research is a non-trivial extension, as we had to …
> 1. Extend BatchBALD to apply to the survival setting, as this extension needs to begin with “partial information” – as the label for Patient X is NOT “no information”, but is “at least 5 years”
> 2.   Extend BatchBALD to provide “incremental information” – perhaps stating that the new label for Patient X should be “at least 6 years”.
> 3. We apply Khuller et al to use the BatchBALD ideas to select the instances to (partially) label, as we reduce our problem to the budgeted maximum coverage problem. (Note this reduction is our work and is novel.) As Khuller et al have already given the optimal solution for this setting,  we use their method. We also needed to alter Khuller et al to be computationally efficient – something we failed to discuss sufficiently in the initial submission,  but is better described in the revision.
>
> Thus, while our method is indeed based on BatchBALD and Khuller et al., we needed to provide many extensions for our system to  work in our “increment”  budgeted learning survival settings.

---

### Official Review · Reviewer_PdWR · 2024-11-04

**Soundness:** 2
**Presentation:** 1
**Contribution:** 2
**Rating:** 1
**Confidence:** 4

**Summary:**

The authors consider the learning of survival model with budget constrained active learning. The formulation of the problem (Section 3) requires that the training dataset has the complete information of event times and censoring times, which is not a typical survival problem in which the survival times cannot be all observed for training data. The active learning is only applied to the case where the budget is used to extend the study (expend the observation period) in order to observed possible more events. This scenario is not the typical consideration in survival analysis in medical or health studies, though in reliability study in industry, in which type II or Type I right censoring are often used to collect data, it is possible to observe more events by increasing the study time.  In clinical trial random right censoring is mostly the observational scheme and  extending study time may not be able to "acquire" the information assumed in the article.

**Strengths:**

The use of active learning in survival analysis is an interesting idea.

**Weaknesses:**

The use of de-censoring as the label in survival analysis does not have much practical usage. It could be used in "reliability" in industry, though. The written English is not clear. The use of words and punctuation like  "and" and "," make the sentences hard to understand. For example, see  lines 269 and 282.

**Questions:**

1. The requirement of true times as well as censoring times for all the subjects in the training data is very restrictive in survival analysis. You may argue that in active learning you just use "complete data" as the initial step, then the  observations are biased, and the model learned from the data does not reflect the true survival model, and thus any further results based on the model are problematic.

2. It seems y denotes the true survival time, but the test data have {x_Bi, y_Bi}, but not ctime_Bi cevent_Bi. It is not clear which are known and which are unknown. The English is hard to understand, too. For example, the meaning of "the model sees only when the data points are uncensored" (line 140, page 3).

3. Survival time is often a continuous random variable, in the defined model (or procedure) in line 160, the previous defined time y is assumed to take values from {1, ..., t}. It is not clear what the notations y and t stand for.

4. The notation "I" has multiple meaning in the context. I-oracle (line 156),  unit of time (line 158, ctime_i +I), and mutual information (line 197). The other notations like the second term on the right hand side of line 200 with the one one the left side of line 207, what the w_j on the right hand of line 207 stands for, and why we have the summation of of j from 1 to k here. You may not discard or add terms arbitrarily to confuse the readers.

5. The BatchBALD method was developed in Kirsch et al (2019) already. It is not clear the information entropy in line 212 is from Kirsch et al (2019) or proposed by the author.

6. For survival data, the authors  tried to obtain the probabilities p(y|x, w D_train) by assigning zero for those value y which below the censoring time c_b, then normalized the probabilities of those values larger then c_b (conditional on the survival time being larger than c_b). This is obvious from the conditional probability rule. From line 297, if the instance is censored at ctime_b, why p(y|x,w,D_train)=0 for all y greater than ctime_i+I. How is the interpretation of the formula in line 301?

---

> ### Author Response · Authors · 2024-11-23
> **Reply to reviewer**
>
> Thank you for your excellent review. Based on your insights, we made a number of changes, to address the weaknesses mentioned. Summary:
> For the summary section, we wish to clarify some points we conveyed poorly:
> The notation in Section 3 suggests that the training dataset has the actual event time for each instance.  The revised version will explicitly note that this is only used by oracle, as we run our simulation – to allow our budgeted-learner to acquire the event time, for the specific instances selected. At any time, our learner will *only* see those event times – for the others, it will only see the censored times.
> In our new draft we discuss a larger scope of where our methods can be used, including reliability testing. (Thanks for the great suggestion!)
>
> Weaknesses:
>
> We have included a new paragraph regarding the scope of the work, and we will continue to improve the writing.
>
> Questions:
>
> 1. Would you mind clarifying the question here? Was this due to  our problematic notation (mentioned above), which suggests that the learner sees all the event times for all instances. Note that the learner will only see the event times provided initially, and those it purchases;  it will see only the censored times for the remaining instances. The revised version corrects this notation.
>
> 2. We have attempted to address this confusion in the new draft.
>
> 3. We use time bins to divide time into t classes, and the random variable y represents one of these classes. This is clearer in the revised text.
>
> 4. We have changed the notations for the increment “I” so that it is different. We apologize for this confusion and will explicitly define the terms in our new draft.
>
> 5. The introductory sections in the revised version  will describe our contributions more clearly.
>
> 6. The new Section 5 expands and clarifies this point.
>
> Please let us know if there are any other parts we should revise.
> Thank you for your review.

---

> > ### Comment · Reviewer_PdWR · 2024-11-25
> > **Response to the rebuttal**
> >
> > I appreciate the authors' responses; however, I found them lacking in detail. As a researcher in survival analysis, I remain unconvinced about the broader applicability of the de-censoring issue, particularly in health and medical fields, which are the primary motivation for its consideration.

---

> > > ### Author Response · Authors · 2024-11-26
> > > **Reply to Reviewer**
> > >
> > > Thank you for your response.
> > >
> > > The revised version mentions other areas (such as reliability testing) where our work can be applied. Within the field of health and medicine specifically, it is easy to motivate wanting to track down subjects who have moved out, and also to continue tracking subjects, after the initial end of the study, as this additional information can help the researchers understand more about a medical issue.  There are also many other examples – eg, after a 6 month study to learn a model that predicts how long a specific type of part will operate, we may want to continue to monitor those “still functional after 6 months” parts, after those 6 months.  Similarly, for customer churn: many consumers may still be shopping at store S after 1 year… we may want to contact them after 2 years, to see if they are still “loyal”. Etc etc etc.

---

### Official Review · Reviewer_APZ6 · 2024-11-04

**Soundness:** 2
**Presentation:** 2
**Contribution:** 2
**Rating:** 1
**Confidence:** 4

**Summary:**

- Presented work focuses on the problem of Bayesian active learning with the learner trained on survival analysis data.
- The number of query steps is constrained by the given budget. In particular:
   - the underlying data are right censored with the underlying latent classifier which indicates whether the patient died during the observation period;
   - the censoring corresponds to a patient dropping out of study and so after a particular time period we lose the access to the information whether the patient survived, the censoring bound is known.
   - the learner can query an unlabeled dataset and obtain partially labelled censored instances, where some time interval can be queried, e.g. data are censored up to 3 years and we query information about additional two more years about the patient.  This corresponds to following up with the patient after some period;
   - the budget constraint indicates that different labels have associated different costs and information content.

Authors propose a solution based on the Bayesian Active learning by Disagreement (BALD) applied to querying a batch of given size.  Authors provide a budgeted greedy algorithm and incorporate mutual information scores from the BALD-like estimate. To budget constrain problem, authors try to optimize the cost coverage over the batch.

Authors use 3 real life survival datasets to demonstrate their proposed method.

**Strengths:**

- The selected problem is very relevant and interesting for real life data applications. The paper well states the problem and aspire to provide a solution.
- Authors propose three alternative methods to demonstrate the strength of their approach.

**Weaknesses:**

1) Paper contains multiple typos, see e.g line 210.
2) Terms are not clearly defined, e.g. Algorithm 1, line 275, what is a_BB_surv ?
4) The paper is not very clearly written, there is missing description of  the learner, there is missing debate on the distributional properties of the BALD model.
5) The results in the main section seems cherry-picked, e.g. it would be good to include more points in the budget constraints rather than 0, 1, 5, 20.
6) It would be good to provide study on the synthetic dataset to demonstrate how the proposed methods perform in the controlled setting.

**Questions:**

1) Is there a typo on the lines 211 - 215? In particular, can authors elaborate in more detail why the second equality ( between lines 212-213) holds?
2) What does authors mean by statement: “ by omitting the first two lines we can reduce the computational complexity …” (lines 252 - 253)
3) Can authors provide more insights into how the prior and likelihood within the BALD are selected?
4) Can you clarify how the proposed “approach provides bounds and time complexity are asymptomatically equivalent to standard active learning methods”? (Lines 29-31).

---

> ### Author Response · Authors · 2024-11-23
> **Reply to reviewer**
>
> Thank you for your review. Based on your insights we have made a number of changes and have addressed the weaknesses mentioned:
>
> 1. We fixed the typo on that line and will continue to reread for typos.
>
> 2. Although we had mentioned  a_BB_surv a few lines before, you are right that we should have expanded on this.  a_BB_surv  is the acquisition function that computes the mutual information between a batch of points and the model parameters, similar to a_BatchBALD for the BatchBALD method, but our acquisition function accounts for partial information and censored data. The revised version will provide this important information.
>
> 3. We apologize for the lack of clarity in writing; the revised version will be more clear. In response to your concerns, what additional information about the learner do you believe we should include? The Bayesian model we used is basically from another paper, with minimal changes.
>  "There is missing debate on the distributional properties of the BALD model" - The BALD model simply takes in a batch of instances and the probability of classes and returns a mutual information measure; what distribution properties are you refering to?
>
> 4. We are currently working on generating more results to address this concern.
>
> 5. We are currently producing additional visuals to more clearly show our point for this concern.
>
> Questions:
>
> 1. The revised version will explicitly note that this equality holds here because the { y_i } variables are independent when conditioned on ω.  Note we did not emphasize the details too much as they are not our work, but appear explicitly in the BatchBALD paper; we will provide more such pointers to formulas and other details that appear in that paper.
>
> 2. The revised paper will include a fundamentally changed statement. The budgeted greedy algorithm includes two lines at the top  that make the algorithm very computationally expensive. We remove these lines for our method. However we recognize expressing it like this is a poor way to illustrate the change, Our revised paper will explicitly demonstrate how we have modified the algorithm
>
> 3. The prior and likelihood of the Bayesian model are selected based on the settings used in the Qi et al paper, which uses these priors on the exact same datasets we used.
>
> 4. We have changed this to simply say "are asymptotically equivalent to the BatchBALD" algorithm, which is a standard active learning model.
>
> Thank you for your review and time.

---

> ### Comment · Reviewer_APZ6 · 2024-11-27
>
> I would like to thank authors for revising their manustript and addressing my comments and questions.
>  - add Question 1: I would like to see the derivation, e.g. included in the Appendix.
> -  add Question 3: I would like to see detailed description of the model, discussion of the distributional properties, discussion related to algorithmic aspects related to computation of the information quantities, posterior, etc. Taking off-the shelf solution from another paper is ok, but given that it is core for the computation of the BALD quantitity, I found unsatisfactory to be given reference to the paper only.
>
> In general the provided responses are rather vague and do not address the points. The responses and the content of the paper misses  the required technical details for ICLR conference. I have decided to change my score to 1.
>
> The problem is relevant and interesting and I encourage authors to go into more depth of the methods they use.

---

> ### Author Response · Authors · 2024-11-30
> **Reply to Reviewer**
>
> Thank you for your review. In hopes of making our paper better, and learning more about the limitations in our presentation we wanted to ask some follow up questions.
>
> 1. Since this entire derivation is the work of the BatchBALD paper, would you recommend we input the derivation required into the appendix verbatim?
>
> 2. For this question, we could discuss the distributional properties of the prior, and how the posterior are computed. What more would you like to know about the algorithmic aspects of BatchBALD beyond what we can provided in 1? we can also include the time and space complexity. Do you think it is ok if we include these in the appendix or should they be included in the main text?
>
> 3. We were wondering if you have any additional recommendations for us to improve this paper?
>
> Thank you again for your time and very helpful insights.

---

### Author Response · Authors · 2024-11-26
**New Draft has been Uploaded.**

Dear Reviewers,

We have uploaded the latest draft of our work. To make the revisions explicit, we have assigned a specific color to each of you, clearly indicating how we addressed your concerns. In sections where the changes are not specific to any particular reviewer, we denote the revisions under the author "m" using orange highlights.

Most of the paper has been updated, except for the results section, where we are awaiting the outcomes of additional experiments. These will be included in the next draft. For the final submission, we will remove the highlights and ensure the manuscript is reduced to the 10-page limit.

We would greatly appreciate your feedback on this draft and whether it has resolved some of your concerns with the paper.

Thank you for your time and effort.

---

### Meta-Review · Area_Chair_Di9Q · 2024-12-21

**Metareview:**

The paper concerns active learning under budget constrains for censored survival data. The Authors adapt the Bayesian Active learning by Disagreement (BALD) to the consider setting. The approach is illustrated using three real life survival problems.

The Reviewers pointed out several issues of the submission such as unclear writing with many typos and inconsistencies in notation, impractical motivation of the studied problem, and limited/unclear novelty. The paper cannot be accepted for publication.

**Additional Comments On Reviewer Discussion:**

The Authors acknowledged many of the critical comments provided by the Reviewers and tried to improve the paper. However, this did not alter the overall negative assessment of the paper. On the contrary, the most positive Reviewer lowered the rating after reading the other reviews and the rebuttal.

---

### Decision · Program_Chairs · 2025-01-22

Reject